# The evolution of colistin resistance increases bacterial resistance to host antimicrobial peptides and virulence

Pramod K Jangir[1]*[†], Lois Ogunlana[1], Petra Szili[2,3], Marton Czikkely[2], Liam P Shaw[1], Emily J Stevens[1], Yang Yu[4], Qiue Yang[5], Yang Wang[6], Csaba Pál[2], Timothy R Walsh[1], Craig R MacLean[1]*

[1]Department of Biology, University of Oxford, Oxford, United Kingdom; [2]Synthetic and Systems Biology Unit, Biological Research Centre, Eötvös Loránd Research Network, Szeged, Hungary; [3]Doctoral School of Multidisciplinary Medical Sciences, University of Szeged, Szeged, Hungary; [4]Guangdong Provincial Key Laboratory of Veterinary Pharmaceutics Development and Safety Evaluation, South China Agricultural University, Guangzhou, China; [5]Fujian Provincial Key Laboratory of Soil Environmental Health and RegulaWon, College of Resources and Environment, Fujian Agriculture and Forestry University, Fuzhou, China; [6]Beijing Advanced Innovation Centre for Food Nutrition and Human Health, College of Veterinary Medicine, China Agricultural University, Beijing, China

*For correspondence:
jangirk.pramod@gmail.com
(PKJ);
craig.maclean@biology.ox.ac.uk
(CRMacL)

Present address: [†]Biozentrum, University of Basel, Basel, Switzerland

Competing interest: The authors declare that no competing interests exist.

**Abstract** Antimicrobial peptides (AMPs) offer a promising solution to the antibiotic resistance crisis. However, an unresolved serious concern is that the evolution of resistance to therapeutic AMPs may generate cross-resistance to host AMPs, compromising a cornerstone of the innate immune response. We systematically tested this hypothesis using globally disseminated mobile colistin resistance (MCR) that has been selected by the use of colistin in agriculture and medicine. Here, we show that MCR provides a selective advantage to *Escherichia coli* in the presence of key AMPs from humans and agricultural animals by increasing AMP resistance. Moreover, MCR promotes bacterial growth in human serum and increases virulence in a *Galleria mellonella* infection model. Our study shows how the anthropogenic use of AMPs can drive the accidental evolution of resistance to the innate immune system of humans and animals. These findings have major implications for the design and use of therapeutic AMPs and suggest that MCR may be difficult to eradicate, even if colistin use is withdrawn.

## Editor's evaluation

Antimicrobial peptides (AMPs) are a class of antibiotics that are inspired by natural components of innate immunity, which raises the specter of bacteria becoming resistant to both. Jangir et al. test this idea and find compelling evidence that a plasmid that encodes resistance to the AMP colistin also increases resistance to AMPs produced by humans, pigs, and chickens, enables the bacteria to grow better in low levels of AMP, and increases bacterial virulence in an insect model of infection. This important study will be of interest to both evolutionary biologists and microbiologists focused on antimicrobial therapy and suggests that the evolution of resistance to these compounds can have collateral effects on immune evasion as well.

## Introduction

The spread of antibiotic resistance in pathogenic bacteria has created an urgent need to develop novel antimicrobials to treat drug-resistant infections. Antimicrobial peptides (AMPs) are multifunctional molecules found among all kingdoms of life that act as key components of the innate immune system of metazoans by modulating immune responses and defending against invading pathogens (*Zasloff, 2002a*; *Yeung et al., 2011*; *Mookherjee et al., 2020*). AMPs are potent antimicrobials with desirable pharmacodynamic properties and a low rate of resistance evolution (*Yu et al., 2018*; *Lazzaro et al., 2020*; *Spohn et al., 2019*; *Jangir et al., 2021*). Given these benefits, there is widespread interest in the development of natural and synthetic AMPs for therapeutic use (*Mookherjee et al., 2020*; *Hancock and Sahl, 2006*; *Magana et al., 2020*). However, a serious concern with the therapeutic use of AMPs is that they share common physicochemical properties and mechanisms of action with AMPs of host immune system, suggesting that the evolution of bacterial resistance to therapeutic AMPs may generate cross-resistance to host AMPs (*Habets and Brockhurst, 2012*; *Kubicek-Sutherland et al., 2017*; *Fleitas and Franco, 2016*; *Andersson et al., 2016*; *Napier et al., 2013*; *Dobias et al., 2017*). Given that host AMPs play important roles in mediating bacterial colonization and fighting infection (*Salzman et al., 2010*; *Ostaff et al., 2013*), cross-resistance to host AMPs could increase pathogen transmission and virulence (*Groisman et al., 1992*; *Kidd et al., 2017*).

Evolutionary microbiologists typically study the consequences of selection for antimicrobial resistance using experimental evolution. In this approach, the pleiotropic responses of bacterial populations that have been selected for increased resistance to an antimicrobial are compared with the responses of unselected control populations (*Pál et al., 2015*; *Imamovic and Sommer, 2013*; *Barbosa et al., 2019*). This is a powerful and tractable approach that has provided important insights into cross-resistance and collateral sensitivity, but the weakness of this approach is that the mechanisms of resistance evolution in the lab do not always match with what occurs in pathogen populations. For example, the evolution of resistance to antibiotics in many pathogens has been largely driven by the acquisition of resistance genes via horizontal gene transfer (*Partridge et al., 2018*; *MacLean and San Millan, 2019*), but conventional experimental evolution approaches focus on variation that is generated by spontaneous mutation. In this article, we use a different approach that is based on testing the pleiotropic impacts of mobile colistin resistance (MCR) genes that have become widely distributed in *Escherichia coli* due to selection mediated by the anthropogenic use of colistin in agriculture and medicine.

Colistin (polymyxin E) is an AMP produced by *Bacillus polymyxa* with similar physicochemical properties and mechanisms of action to metazoan AMPs (*Rodríguez-Rojas et al., 2015*; *Vaara, 1992*; *Supplementary file 1*). Colistin began to be used at a large scale in agriculture in the 1980s (*Wang et al., 2017*), but it is being increasingly used as a 'last-resort' antimicrobial to treat infections caused by multidrug-resistant (MDR) Gram-negative pathogens (*Li et al., 2006*). Colistin resistance has evolved in many pathogens (*Jochumsen et al., 2016*; *Sun et al., 2020*; *Snitkin et al., 2013*), but the most concerning case of colistin resistance evolution comes from MCR genes in *E. coli*, as exemplified by *mcr-1* (*Liu et al., 2016*). Phylogenetic analyses show that *E. coli* acquired a composite transposon carrying *mcr-1* in China at some point in the 2000s (*Wang et al., 2018*). MCR initially spread in populations of *E. coli* from farms, where colistin was used as a growth promoter to increase the yield of chicken and pig production. However, *mcr-1* became widely distributed across agricultural, human, and environmental sources due to the combined effects of bacterial migration and rapid horizontal transfer of *mcr-1* between plasmid replicons and host strains of *E. coli* (*Wang et al., 2017*; *Sun et al., 2018*; *Gao et al., 2016*).

MCR-1 transfers phosphoethanolamine (pEtN) to lipid-A in the cell membrane, resulting in decreased net negative cell surface charge and thus lower affinity to positively charged colistin (*Sun et al., 2018*). Crucially, loss of cell surface charge through membrane modification is a common resistance mechanism against cationic AMPs across bacteria (*Spohn et al., 2019*; *Andersson et al., 2016*), suggesting that MCR-1 may provide cross-resistance to host AMPs. However, membrane alterations produced by MCR-1 expression are associated with clear costs (*Yang et al., 2017*), and it is equally possible that membrane remodeling could generate collateral sensitivity to AMPs, as has been observed with antibiotic resistance genes (*Lázár et al., 2018*).

In this article, we test the hypothesis that evolving colistin resistance via MCR gene acquisition provides bacteria with increased resistance to host AMPs. *mcr-1* is usually carried on conjugative

**Table 1.** List of natural mobile colistin resistance (MCR) plasmids and antimicrobial peptides (AMPs) used in this study.

**AMPs**

| Name | Abbreviation | Major cell and tissue sources | |
|---|---|---|---|
| LL-37 cathelicidin | LL37 | Epithelial cells of the testis, skin, gastrointestinal tract, respiratory tract, and in leukocytes, such as monocytes, neutrophils, T cells, NK cells, and B cells | |
| Human beta-defensin-3 | HBD3 | Neutrophils and epithelial surfaces (e.g., skin, oral, mammary, lung, urinary, eccrine ducts, and ocular) | |
| Cecropin P1* | CP1 | Small intestine | |
| PR39 | PR39 | Mucosa and lymphatic tissue of the respiratory tract | |
| Protegrin 1 | PRO1 | Bone marrow, leukocytes, and neutrophils | |
| Prophenin-1 | PROPH | Bone marrow and leukocytes | |
| PMAP-23 | PMAP23 | Myeloid tissue, bone marrow, and liver | |
| Chicken cathelicidin-2 | CATH2 | Bone marrow, respiratory tract, gastrointestinal tract, normal intact skin, and multiple lymphoid organs | |
| Fowlicidin 3 | FOW3 | Bone marrow, lung, and spleen | |
| Colistin | COL | - | |

**MCR plasmids**

| Name (type) | Size (bp) | mcr gene | Reference |
|---|---|---|---|
| PN16 (IncI2) | 60,488 | mcr-1 | *Yang et al., 2017* |
| PN21 (IncI2) | 60,989 | mcr-1 | *Yang et al., 2017* |
| PN23 (IncX4) | 33,858 | mcr-1 | *Yang et al., 2017* |
| PN42 (IncX4) | 32,995 | mcr-1 | *Yang et al., 2017* |
| WJ1 (IncHI2) | 261,119 | mcr-3 | *Yin, 2017* |
| 481 (IncP1) | 53,660 | mcr-3 | *Wang et al., 2019* |

*From pig intestinal parasitic nematode.

plasmids from a diversity of plasmid incompatibility types (such as IncX4, IncI2, IncHI2, and IncP1) that carry a large number of housekeeping and cargo genes (*Wang et al., 2018*; *Yang et al., 2017*). We assessed the importance of this diversity by transferring a diversity of naturally occurring plasmids and synthetic MCR-1 expression vectors to a single recipient strain of *E. coli*. To assess the impact of MCR on resistance to host AMPs, we screened a panel of strains carrying naturally occurring and synthetic MCR plasmids against a collection of AMPs. Given the importance of agricultural animals as reservoirs of *mcr-1,* we tested AMPs that play important roles in the innate immunity of humans, pigs, and chickens (*Table 1*). Next, we examined the role of MCR-1 in complex host environments and bacterial virulence using human serum resistance assays and in vivo virulence assays in the *Galleria mellonella* infection model system. The key innovation in this study is that we have taken a systematic approach to testing the pleiotropic effects of the dominant mechanism of colistin resistance evolution, including assessing the impact of AMP resistance on bacterial virulence.

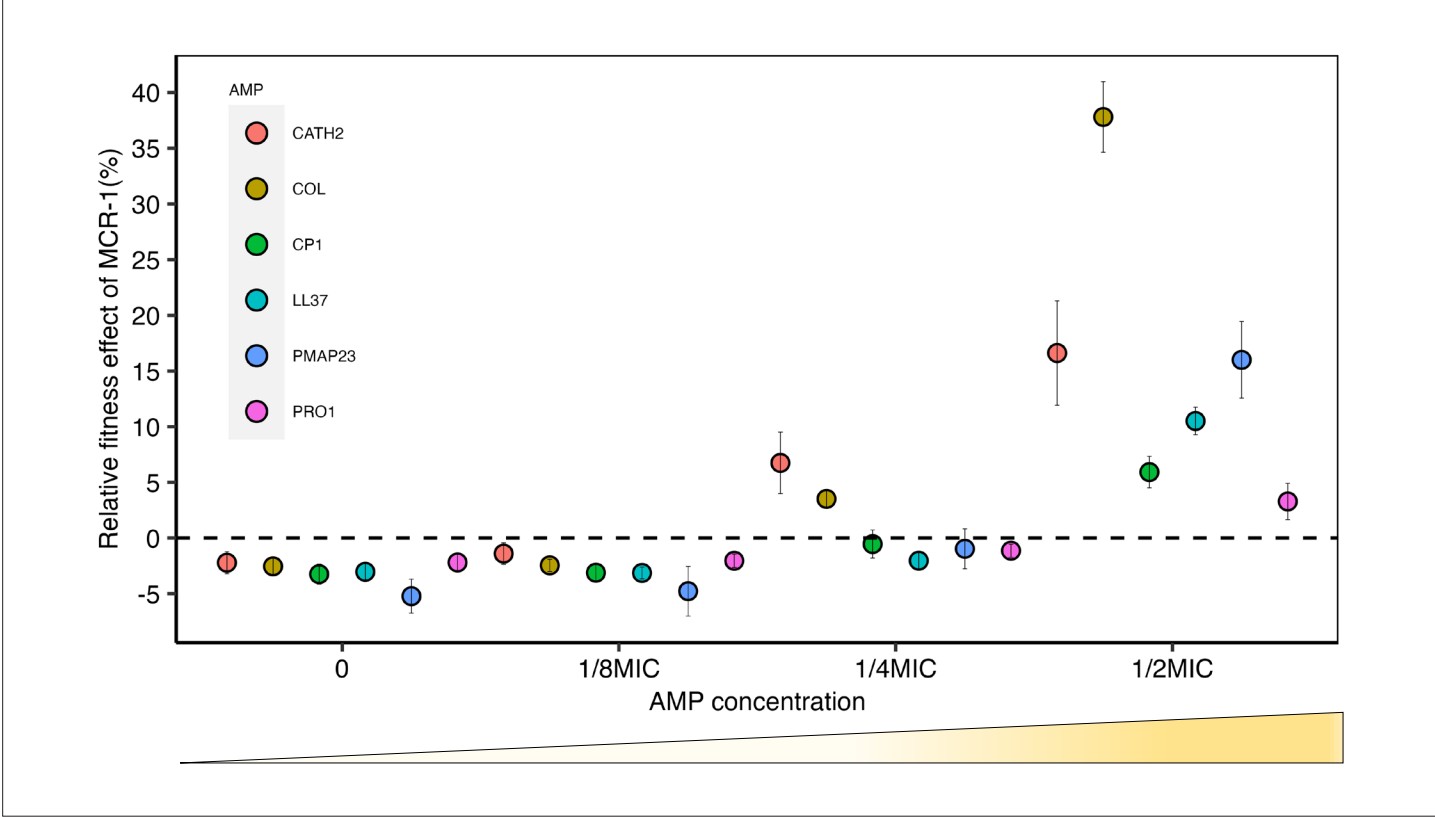

**Figure 1.** Sub-minimum inhibitory concentration (sub-MIC) doses of antimicrobial peptides (AMPs) generate selection for mobile colistin resistance (MCR). *E. coli* carrying *mcr-1* expression vector (pSEVA:MCR-1) or an empty vector control (pSEVA:EV) were competed against a tester strain carrying a chromosomally integrated GFP across a range of AMP concentrations (n = 6 biological replicates per competition). Plotted points show the competitive fitness effect of the MCR-1 expressing strain relative to the empty vector control (±SE). To facilitate comparisons across AMPs, fitness is plotted as a function of relative AMP concentration, and the dashed line represents equal fitness.

The online version of this article includes the following figure supplement(s) for figure 1:

**Figure supplement 1.** MCR-1 imposes a significant fitness burden in the absence of an antimicrobial peptide (AMP) (p=1.174e-15, from two-sided Mann–Whitney *U*-test, n = 36 for each genotype).

**Figure supplement 2.** Gating strategy to analyze the selective fitness benefits of MCR-1.

## Results

### Host AMPs select for MCR-1

To assess the consequences of MCR acquisition without any confounding effects from backbone and cargo genes found in naturally occurring MCR plasmids, we cloned *mcr-1* and its promoter into a non-conjugative expression vector (pSEVA121) that has a similar copy number to naturally occurring MCR plasmids (approximately five copies per cell). As a first approach to assess the impact of MCR-1 on resistance to host AMPs, we measured the competitive ability of pSEVA:MCR-1 across a concentration gradient of a randomly selected representative set of host AMPs and colistin, which acts as a positive control for MCR selection (*Figure 1*).

Consistent with previous work (*Yang et al., 2017*), *mcr-1* imposed a significant fitness burden in the absence of AMPs, reducing competitive ability by 3% (p=1.174e-15, two-sided Mann–Whitney *U*-test, *Figure 1—figure supplement 1*). However, *mcr-1* provided a significant competitive fitness advantage at concentrations of host AMPs between ¼ and ½ of minimum inhibitory concentration (MIC) (*Figure 1*, *Supplementary file 2*). Although *mcr-1* provided a greater fitness advantage in the presence of colistin as compared to host AMPs, the minimal selective concentration for colistin, ¼ MIC, was only marginally lower (*Figure 1*). It is important to note that the sub-MICs required for the selection of *mcr-1* overlap with the range of physiological concentration of host AMPs. For example,

the concentration of LL-37 required to select for MCR-1 (~3.4 µM) falls well within the reported physiological concentration range (up to 10 µM) (*Barlow et al., 2010*; *Srakaew et al., 2014*).

## MCR increases resistance to host defense AMPs

To test the hypothesis that MCR increases resistance to host AMPs more broadly, we measured the resistance of MCR-*E. coli* to a panel of AMPs. Given the importance of agricultural animals as reservoirs of MCR (*Liu et al., 2016*), we tested AMPs that are known to play important roles in the innate immunity of chickens, pigs, and humans. The panel of AMPs used in our assay have diverse mechanistic and physicochemical properties (*Supplementary file 1*) and include AMPs that are known to have clinical relevance and play key roles in mediating innate immunity (*Table 1* and *Supplementary file 1*). For example, the human cathelicidin LL-37 and defensin HBD-3 have immunomodulatory activities in addition to their antimicrobial activity (*Mookherjee et al., 2020*; *Zasloff, 2002b*; *Zhang and Gallo, 2016*).

We tested the AMP resistance of both *E. coli* carrying pSEVA:MCR-1, which provides a clean test for the effect of the *mcr* gene, and transconjugants carrying diverse *mcr-1* and *mcr*-3 natural plasmids. These plasmids represent the dominant platforms for MCR found in clinical and agricultural sources in Southeast Asia (*Wang et al., 2018*), and plasmid diversity may play an important role in mediating the effect of MCR due to variation in plasmid copy number and the effect of other plasmid genes on AMP resistance.

One key difference between resistance to AMPs and antibiotics is that AMP resistance genes typically give much smaller increases in resistance than antibiotic resistance genes, typically on the order of one- to twofold increases in MIC (*Kintses et al., 2019*). No standardized methods exist to measure resistance to AMPs, and we measured AMP resistance using an established assay that had the sensitivity to capture small differences in bacterial resistance that are missed by conventional antibiotic resistance assays (i.e., less than twofold changes in MIC) (*Kintses et al., 2019*; *Figure 2*).

The significance of changes in antibiotic resistance is usually determined by comparing the MIC of strains carrying a resistance gene to established clinical breakpoints. No such breakpoints exist for AMP resistance, and we tested for statistically significant changes in resistance to AMPs associated with MCR. On average, MCR plasmids provided increased resistance to host AMPs by 62% (mean fold change in MIC = 1.62; SEM = 0.11; t = 5.615; p<0.0001; *Figure 2*; *Supplementary file 3*). However, the average change in resistance conferred by MCR plasmids varied significantly between AMPs as MCR plasmids increased resistance to most AMPs, but generated collateral sensitivities to both PROPH and PR39 (*Figure 2b*; $F_{8,40}$ = 7.85; p<0.0001). Our AMP resistance assay did not use standardized culture media that are used to assess antibiotic resistance (i.e., cation-adjusted MHBII) as AMPs act differently than antibiotics and there are no established methodologies to measuring resistance. However, carrying out a subset of AMP resistance assays in standardized media recovered the key result of our assay – that MCR increases AMP resistance (*Figure 2—figure supplement 1*).

The AMP resistance profile of natural MCR plasmids was highly correlated with that of the synthetic pSEVA:MCR-1 expression vector, suggesting that changes in resistance observed in MCR plasmids were caused by MCR, and not by other genes present on these plasmids ($r^2$ = 0.845; *Figure 2—figure supplement 2*). To further test this idea, we replaced the *mcr-1* gene on an IncX4 natural plasmid (PN23 IncX4) with an ampicillin resistance marker, which is not known to have any effect on AMP resistance. As expected, deletion of *mcr-1* gene resulted in a wild-type level of resistance to AMPs (*Figure 2—figure supplement 3*). Altogether, these results suggest that the observed AMP resistance phenotype is largely due to the pleiotropic effects of MCR gene and is not distorted by other genes present on natural plasmids.

MCR generated large increases in resistance to colistin compared to host AMPs, supporting the idea that MCR genes are specialized for providing colistin resistance. This difference is striking given that colistin shares some clear similarities with some membrane-targeting host AMPs in terms of biophysical properties that shape modes of action, such as charge, alpha helix, and aliphatic index (*Figure 2—figure supplement 4*; *Napier et al., 2013*; *Lázár et al., 2018*; *Kintses et al., 2019*). Although there are some shared structural features found in multiple AMPs, they are clearly diverse at a biophysical level. Interestingly, MCR generated collateral sensitivities to both PROPH and PR39, even if there was no change in susceptibility on average (*Figure 2b*). These AMPs have unique physicochemical properties, including high proline content (*Figure 2—figure supplement 4*), which has

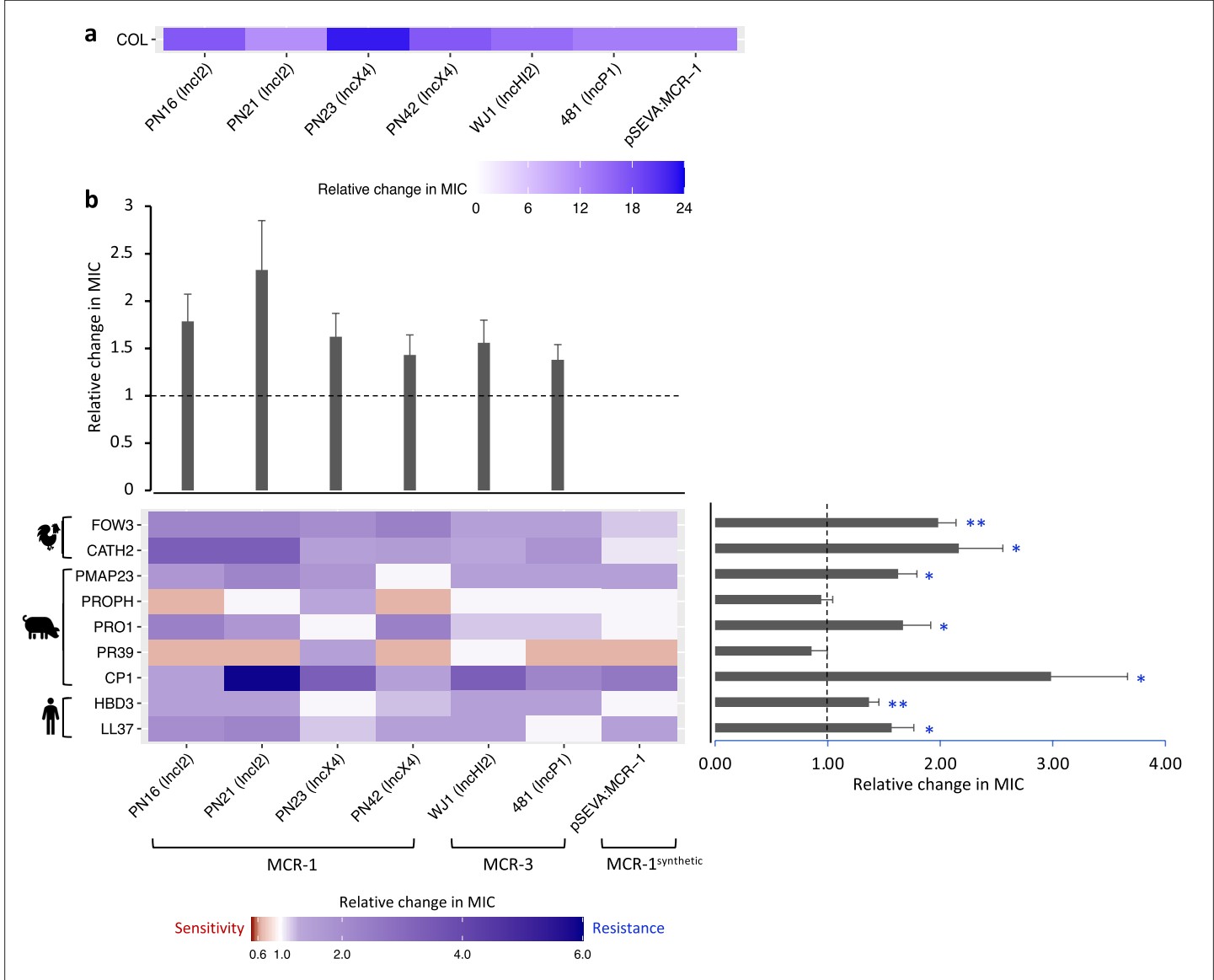

**Figure 2.** Mobile colistin resistance (MCR)-mediated changes in bacterial susceptibility to host antimicrobial peptides (AMPs). Heatmaps depict the effect of MCR plasmids on resistance to colistin (**a**) and host AMPs (**b**). Bacterial susceptibility to AMPs was tested by measuring minimum inhibitory concentrations (MICs), and changes in resistance were assessed relative to control strains lacking MCR (n = 3 biological replicates per MIC). Natural plasmids carried either MCR-1 or MCR-3 are shown according to plasmid incompatibility group. Resistance for these plasmids was measured relative to the *E. coli* J53 parental strain. The impact of the synthetic pSEVA:MCR-1 plasmid on resistance was measured relative to a strain with a pSEVA empty vector. Dashed lines represent control strain. Bar plots show average changes in MIC for natural MCR plasmids and did not include pSEVA:MCR1 (±SE; n = 9 for host AMPs, n = 6 for plasmids; *p<0.05 one-sample *t*-test, LL37- 0.033; HBD3- 0.0088; CP1- 0.032; PR39- 0.353; PRO1- 0.0424; PROPH- 0.5964; PMAP23- 0.0136; CATH2- 0.030; FOW3- 0.001).

The online version of this article includes the following figure supplement(s) for figure 2:

**Figure supplement 1.** Antimicrobial peptide (AMP) susceptibility of *E. coli* carrying mobile colistin resistance (MCR) natural plasmids and the synthetic pSEVA:MCR-1.

**Figure supplement 2.** Antimicrobial peptide (AMP) resistance phenotype obtained for pSEVA:MCR-1 is highly correlated with those from the naturally occurring MCR-1 plasmids (linear regression $F_{1,8}$ = 43.7, p=0.0002, $r^2$ = 0.845).

**Figure supplement 3.** Deletion of MCR-1 from IncX4 natural plasmid results in antimicrobial peptide (AMP) susceptibility similar to the wild-type (WT) control strain.

**Figure supplement 4.** PR39 and PROPH differ from other antimicrobial peptides (AMPs) in their physicochemical properties.

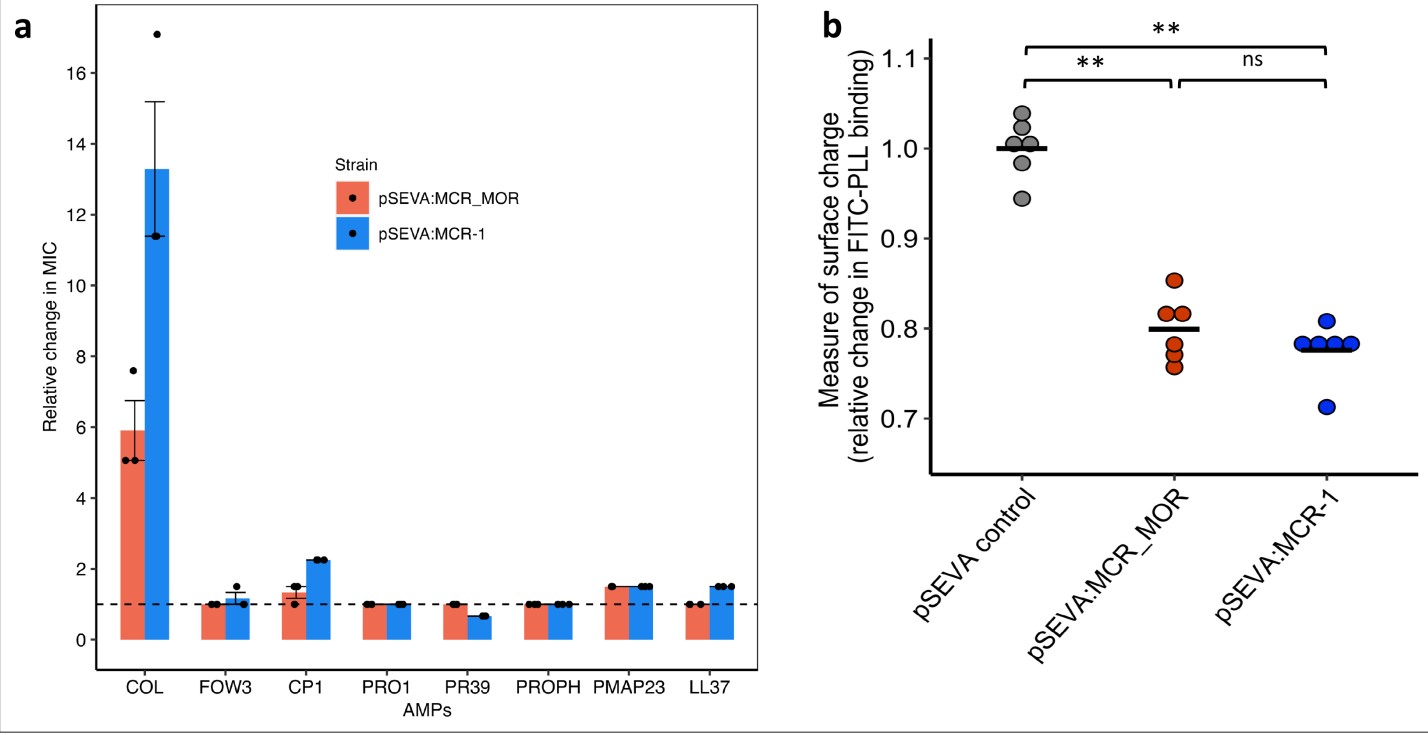

**Figure 3.** Effect of *Moraxella* MCR (MCR-MOR) on bacterial susceptibility to antimicrobial peptides (AMPs) (**a**) and on cell surface charge (**b**). (**a**) AMP susceptibility of *E. coli* carrying pSEVA:MCR-1 and *Moraxella* version of MCR (pSEVA:MCR-MOR). The impact of the pSEVA:MCR-1 and pSEVA:MCR-MOR on resistance was measured relative to a strain with a pSEVA empty vector control (dashed line). Error bars indicate standard errors based on three biological replicates. (**b**) Relative cell surface charge of *E. coli* strains expressing MCR-1 and MCR-MOR compared to an empty vector control. Cell surface was determined by FITC-PLL binding assay (n = 6 biological replicates/strain). Statistical significance was determined by pairwise comparisons using the two-sided Mann–Whitney *U*-tests, and double asterisks show differences with a p-value<0.01.

been shown to be a common property of intracellular-targeting AMPs, as opposed to membrane-disrupting AMPs (*Supplementary file 1*; *Scocchi et al., 2011*; *Gerstel et al., 2018*).

To better understand the origins of the high colistin resistance phenotype associated with *mcr-1*, we cloned the closest known homologue of MCR-1 from the pig commensal *Moraxella* (MCR-MOR, ~62% [amino acid] identity with MCR-1) into pSEVA121 (*Sun et al., 2018*; *Kieffer et al., 2017*). In general, MCR-MOR expression was associated with small changes in susceptibility to AMPs compared to MCR-1 (*Figure 3a*). In line with previous work, MCR-MOR expression provided a small increase (5.9-fold) in colistin resistance compared to MCR-1 (13.2-fold) (*Wei et al., 2018*; *AbuOun et al., 2017*; *Figure 3a*).

Loss of negative membrane charge has been argued to play an important role in the colistin resistance provided by *mcr-1*. MCR-1 is a pEtN transferase enzyme that facilitates the addition of pEtN to the lipid A component of lipopolysaccharide (LPS), resulting in reduced binding of colistin. However, MCR-1 and MCR-MOR have similar effects on cell surface charge (*Figure 3b*, p=0.470, two-sided Mann–Whitney *U*-test), supporting the idea that MCR-1-mediated colistin resistance is also attributable to other factors, such as the increased protection of the cytoplasmic membrane from colisitin (*Sabnis et al., 2021*). Given that MCR-MOR does not confer broad resistance to host AMPs, our results suggest that MCR-1 was able to evolve to increase resistance to both colistin and relevant host AMPs.

## MCR confers serum resistance and increases virulence

The above experiments focused on measuring the impact of MCR-1 on bacterial resistance to individual host AMPs. To better understand the protective role of MCR-1 in a complex host environment, we measured bacterial susceptibility to human serum, which contains a complex mixture of antimicrobials, including complement. For this assay, we selected IncI2 and IncX4 plasmids as they are the most

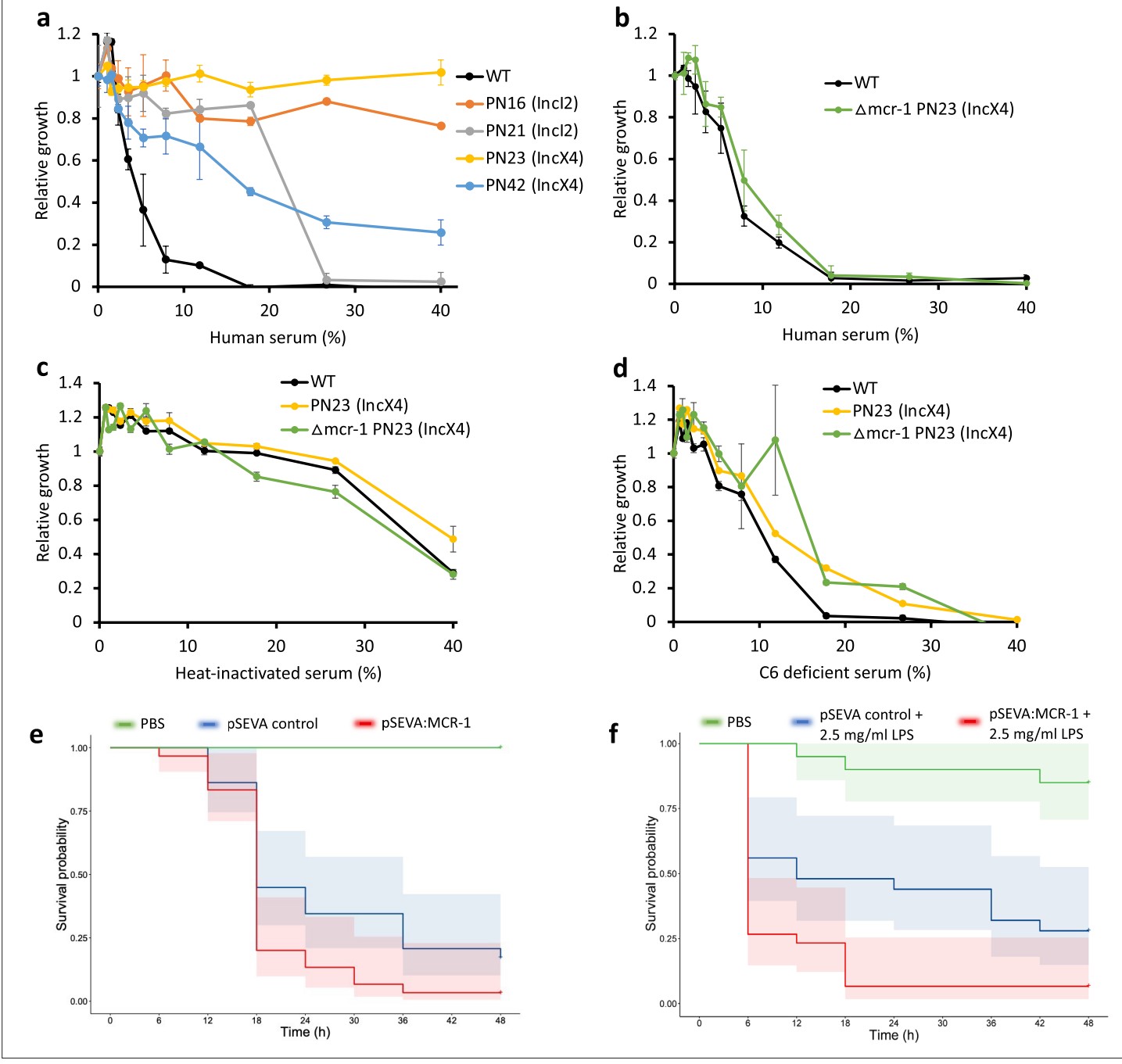

**Figure 4.** Mobile colistin resistance (MCR) confers resistance to human serum and increases bacterial virulence. (**a–d**) Bacterial susceptibility to human serum was determined by measuring bacterial growth in a medium containing human serum relative to serum-free controls ( ± SEM; n = 3 biological replicates/strain). (**a**) Serum susceptibility of the wild-type (WT) parental strain and transconjugants carrying natural MCR plasmids. (**b**) Serum susceptibility of the WT parental strain and transconjugant carrying a plasmid with a deletion of *mcr-1*. (**c, d**) Susceptibility of *E. coli* with and without natural MCR plasmid against C6-deficient and heat-inactivated serum. (**e, f**) Survival of *G. mellonella* larvae following injection with 5 × 10⁷ *E. coli* carrying pSEVA:MCR-1 or an empty vector control compared to mock-treated larvae that were injected with PBS. In (**f**), larvae were pretreated with LPS for 24 hr prior to bacterial infection. Each experiment was performed in triplicate with 10 animals per treatment per replicate, and shaded areas show 95% confidence intervals in survival probability.

dominant MCR-1 plasmid types (*Wang et al., 2018*; *Sun et al., 2018*). Interestingly, these MCR plasmids conferred high levels of resistance to human serum, showing that MCR-1 is effective at providing protection against even complex mixtures of antimicrobials (*Figure 4a*). To rule out that the observed serum resistance is due to MCR-1 and not because of pleiotropic effects of other genes present on the plasmid, we tested serum susceptibility of a strain carrying an MCR-1 knockout IncX4 plasmid. We found no difference in serum resistance between wild-type (carrying no plasmid) and strain with MCR-1 knockout plasmid, suggesting that indeed the observed serum resistance phenotype was due to MCR-1 (*Figure 4b*).

Host complement systems play a major role in bacterial killing by serum, suggesting that MCR-1 may provide resistance against the complement system. To address this, we first tested susceptibility to heat-inactivated serum (*Figure 4c*). Heat inactivation clearly reduced the toxicity of serum to the WT control strain, highlighting the antibacterial effects of heat labile components of serum. All the tested strains showed similar levels of sensitivity to heat-inactivated serum, suggesting that MCR-1-mediated protection against serum (i.e., *Figure 4a and b*) is due to increased resistance to heat-labile effectors that are present in serum. To further probe the role of MCR-1 in providing serum resistance, we then measured growth in serum lacking complement component 6 (C6), a key component of membrane attack complex that induces transmembrane channel and thus lysis of the target bacterial cells (*Figure 4d*). The presence of functional MCR-1 was not associated with increased resistance to C6-defficient serum, providing further evidence to support the role that MCR-1 protects against the complement system.

These results raised the intriguing possibility that increased AMP resistance provided by MCR-1 could increase bacterial virulence by compromising host innate immunity. This is plausible as AMP resistance in pathogens has been shown to be an important virulence factor (*Groisman et al., 1992*). In contrast to this expectation, previous work has shown that MCR-1 plasmids actually decrease virulence in a *G. mellonella* model (*Yang et al., 2017*). However, this study also showed that plasmids with identical *mcr-1* genes had differential effects on virulence, suggesting that these plasmids had effects on virulence that were unrelated to MCR-1. To directly test the impact of MCR-1 on virulence, we measured the impact of the pSEVA:MCR-1 on virulence in the *G. mellonella* infection model. The key advantage of this system is that pSEVA makes it possible to measure the impact of realistic levels of MCR-1 expression, while controlling for any background plasmid effects using an empty vector control. Crucially, the MCR-1 carrying strain showed increased virulence compared to the control strain with an empty vector in spite of the cost associated with MCR-1 expression (*Figure 4e*, log-rank test, p=0.024, *Figure 1—figure supplement 1*).

MCR-1-mediated modification of LPS can result in reduced stimulation of macrophages and limited release of inflammatory molecules, suggesting that MCR-1 could increase virulence by reducing host immunostimulation (*Yang et al., 2017*). If this is the case, then stimulating host immunity should attenuate the effect of MCR-1 on virulence. To test this idea, we measured the impact of MCR-1 on virulence in *G. mellonella* larvae that had been pretreated with LPS, stimulating innate immunity (*Mukherjee et al., 2010*). However, MCR-1 continued to increase virulence in LPS-treated larvae, suggesting that reduced host immunostimulation was not responsible for the increased virulence associated with MCR-1 expression (*Figure 4f*, log-rank test p=0.0074).

## Discussion

AMPs have been advocated as a potential therapeutic solution to the AMR crisis, and colistin resistance provides a unique opportunity to study the evolutionary consequences of large-scale anthropogenic AMP use. Our study shows that MCR increases bacterial fitness and resistance in the presence of AMPs from humans and agricultural animals that act as important sources of MCR carrying *E. coli* (*Figures 1 and 2*). MCR-1 also increases resistance to human serum and virulence in an insect infection model, highlighting the threat of infections caused by MCR-*E. coli* (*Yin et al., 2021*). These findings suggest that MCR-1 provides effective resistance against AMP cocktails that are found in host tissues, but it is important to emphasize that MCR-mediated protection against other antimicrobials, such as lysozyme (*Sherman et al., 2016*) and complement systems (*Figure 4*), may also contribute to this protective phenotype.

Mobile antibiotic resistance genes often confer very large increases in resistance to antibiotics, resulting in a qualitatively different resistant phenotype (i.e., 10- to 100-fold increases in MIC). The

increases in resistance to host AMPs associated with MCR are very modest (typically less than threefold increase) when viewed from an antibiotic resistance perspective. However, these subtle and quantitative changes are entirely consistent with previous studies showing that AMP resistance genes typically confer small changes in resistance (*Kintses et al., 2019*). Our study shows that subtle changes in AMP resistance are associated with clear selective advantages (i.e., >5%) under clinically realistic concentrations of AMPs (*Figure 1*). Moreover, there is growing evidence that mutations leading to small changes in resistance are selected in antibiotic-treated patients (*Frimodt-Møller et al., 2018*; *Wheatley et al., 2021*). Small effect resistance is likely to be particularly important under conditions when multiple resistance mechanisms can be sequentially acquired, generating a high resistance phenotype (*Jochumsen et al., 2016*; *Jangir et al., 2022*; *Papkou et al., 2020*; *Hughes and Andersson, 2017*; *Toprak et al., 2012*).

One of the most important insights from this study is that anthropogenic use of AMPs (e.g., colistin) can inadvertently drive the evolution of resistance to key components of innate immunity (*Habets and Brockhurst, 2012*; *Perron et al., 2006*). Numerous AMPs are currently in clinical trials, including AMPs of human origin (*Mookherjee et al., 2020*; *Lazzaro et al., 2020*), and our results highlight the importance of assessing the impact of evolved resistance to therapeutic AMPs on resistance to host innate immunity and bacterial virulence during preclinical development using sensitive and quantitative assays. It is possible, of course, that resistance to therapeutic AMPs will not be always associated with cross-resistance to host AMPs, as we observed for PROPH and PR39 (*Figure 2b*). However, we argue that cross-resistance to host AMPs is likely to be widespread, given that AMPs tend to share broad cellular targets and physicochemical properties (*Supplementary file 1*). If this is the case, then it is conceivable that mechanisms that have evolved to provide pathogenic bacteria with protection against host AMPs may also help to accelerate the evolution of resistance to therapeutic AMPs (*Jangir et al., 2022*; *Kapel et al., 2022*).

MCR-1 initially spread in agricultural settings in China, where colistin was heavily used as a growth promoter. The Chinese government banned the use of colistin as a growth promoter in 2016, and this was followed by a decline in the prevalence of MCR in human, agricultural, and environmental samples at a national level, providing strong evidence that colistin use in agriculture was the key driver for MCR-1 (*Shen et al., 2020*; *Wang et al., 2020*). The fitness costs associated with MCR-1 (*Yang et al., 2017*) are likely to have played an important role in the decline of colistin resistance, but our findings suggest that AMPs from humans and agricultural animals provide a selective advantage for MCR-1 that has helped to offset the cost of colistin resistance. The doses of AMPs required for MCR-1 resistance selection (~1/2 MIC) are high compared to those that are needed to select for antibiotic resistance (typically <1/10 MIC). However, AMPs achieve high concentrations in host tissues with acute or chronic inflammation (*Mookherjee et al., 2020*; *Fahlgren et al., 2003*), and our results suggest that the selective benefits of AMP resistance may help to maintain MCR-1 in humans and animals, even if colistin usage remains low. It should be noted that, at intracellular sites, the concentration of different AMPs can be extremely high in some instances, and thus, MCR-mediated selective benefits in such conditions remain unclear.

Is MCR-1-mediated evasion of immunity important in clinical settings? Interestingly, the proportion of human infection isolates with MCR-1 remained at a constant low level following the ban on the use of colistin as an agricultural growth promoter (~1–2%) in contrast to healthy human carriage isolates, which went from 21% in 2016 to just 0.8% in 2018 (*Shen et al., 2020*). The proportion of MCR-1-positive isolates that were from the 'pathogenic' phylogroup B2 was much larger in infection isolates compared to healthy carriage (33% vs. 2%). At face value, this might be taken to suggest an association between B2 and MCR-1 in human infection, consistent with our hypothesis of additional selective advantages for MCR-1. However, it is important to control for the population structure of *E. coli*. Infection isolates typically have a greater proportion of the phylogroups B2 and D compared to carriage isolates, with this proportion varying depending on setting. Unfortunately, studies of MCR-1 that use whole-genome sequencing almost invariably sequence only MCR-1-positive isolates. This means that the population structure of the MCR-1-negative isolates in the study remains unknown, so while our estimates from the available data suggest that an association between MCR-1 and B2 in infection may be possible (Appendix 1) we cannot reach a conclusion either way. This limitation highlights the value of understanding the 'denominator' of the wider population structure of clinical pathogens when studying AMR, and it highlights the importance of further understanding the role of MCR in human infection.

Our approach to understanding the consequences of AMP resistance evolution focused on testing the importance of the diversity of plasmid replicons that carry MCR-1. The limitation of this approach is that we tested all of these plasmid types in a single wild-type host strain. This is a limitation because the extensive genetic diversity of *E. coli* ensures that a single strain cannot be assumed to be the representative of this species. An interesting avenue for further work will be to test for epistatic effects of MCR-1 across host strains.

A further limitation of our study is the challenge of understanding the selective benefits of increased resistance to host immunity. We found clear evidence that MCR-1 increases resistance to serum, and this is at least partially attributable to increased resistance to complement and other heat-labile anti-bacterials, such as AMPs and complement systems. However, we were not able to quantitatively assess the fitness benefit provided by MCR-1 in serum, making it difficult to estimate the selective advantage of increased serum resistance and the extent to which this is driven by increased resistance to host AMPs as opposed to other antimicrobial, such as the complement pathway. Similarly, we were not able to measure the selective advantage of MCR-1 in *Galleria* or determine the extent to which increased virulence was driven by decreased susceptibility to insect AMPs in this system. For example, it is possible that increased virulence stems from changes to host tissue invasion and growth stemming from cell membrane alterations mediated by MCR-1, and not increased resistance to host immunity.

## Methods

### Bacterial strains, MCR plasmids, and growth medium

All the experiments were done in *E. coli* strain J53 genetic background. All bacterial strains and MCR plasmids used in this study are listed in *Supplementary file 5* and *Supplementary file 6*. Experiments were conducted in Mueller–Hinton (MH) medium and Luria–Bertani (LB) medium. All components were purchased from Sigma-Aldrich.

### Antimicrobial peptides

AMPs were custom-synthesized by BioServ UK Ltd, except for HBD-3 and colistin. HBD-3 was custom-synthesized by PeptideSynthetics UK, and colistin was purchased from Sigma-Aldrich. AMP solutions were prepared in sterile water and stored at −80°C until further use.

### Oligonucleotides

A full list of DNA oligonucleotides used in this work is provided in *Supplementary file 4*. All oligos were ordered with standard desalting from Thermo Scientific.

### pSEVA:MCR-1 vector construction

A synthetic MCR-1 plasmid was constructed by cloning *mcr-1* gene into pSEVA121 plasmid (*Silva-Rocha et al., 2013*). The *mcr-1* gene along with its natural promoter was PCR-amplified from the natural PN16 (IncI2) plasmid using Q5 High-Fidelity DNA Polymerase (New England BioLabs). The amplified and purified *mcr-1* fragment was cloned into PCR-amplified pSEVA121 backbone using NEBuilder HiFi DNA Assembly Master Mix according to the manufacturer's instructions. Assembled products were then transformed into *E. coli* J53 strain using the standard electroporation method. Briefly, pSEVA121:MCR-1 plasmid-carrying cells were grown overnight in MHB medium supplemented with 100 µg/ml ampicillin. Plasmid DNA isolation was performed using GeneJET Plasmid Miniprep Kit (Thermo Scientific) according to the manufacturer's instructions. 1 µl of the purified plasmid DNA was transformed by electroporation into 50 µl of electrocompetent *E. coli* J53 cells. Electroporation was carried out with a standard protocol for a 1 mm electroporation cuvette. Cells were recovered in 1 ml SOC medium, followed by 1 hr incubation at 37°C. Different dilutions of transformant mixture were made and were plated onto Petri dishes containing LB agar supplemented with 100 µg/ml ampicillin. The culture plates were incubated at 37°C overnight.

PCR and sequence verification by Sanger sequencing were performed to ensure the presence of the correctly assembled recombinant plasmid. A full list of the primers used is given in *Supplementary file 4*.

### Construction of Δ*mcr-1* PN23 (IncX4) plasmid

Gibson assembly was used to construct Δ*mcr-1* PN23 (IncX4) mutant where *mcr-1* gene was replaced by ampicillin resistance marker. The primers used for the Gibson assembly are listed in *Supplementary*

*file 4*. The overlap between fragments to be assembled was in the range of 20–40 bp. To avoid any mutation incorporation in the assembly, Q5 High-Fidelity 2X Master Mix (New England BioLabs) was used for PCR amplification. Five PCR fragments (leaving MCR-1 out) were generated using natural PN23 IncX4 plasmid as template DNA in Q5 High-Fidelity 2X Master Mix with corresponding primer sets (*Supplementary file 4*). An ampicillin resistance marker was amplified separately.

To remove any plasmid DNA template contamination, the amplified PCR products were digested with DpnI (New England BioLabs) for 1 hr at 37°C, followed by 20 min heat inactivation at 80°C. The digested PCR products were subjected to gel purification using GeneJET Gel Extraction and DNA Cleanup Micro Kit (Thermo Scientific). The gel-purified PCR products were assembled together with the ampicillin marker fragment using NEBuilder HiFi DNA Assembly Master Mix according to the manufacturer's instructions. The resulting assembled plasmid DNA was transformed into *E. coli* strain MG1655, rather transforming directly into *E. coli* J53. This extra step was to ensure efficient transformation of the assembled plasmid. *E. coli* MG1655 is a well lab-adapted strain and shows high transformation efficiency, especially for large plasmids. The transformants were selected on LB agar containing ampicillin 100 µg/ml. The presence and right orientation of all six fragments were confirmed by PCR amplification of fragments junction. Similarly, the absence of *mcr-1* gene was also confirmed by PCR. Following the confirmation of the Δ*mcr-1* PN23 (IncX4) plasmid, a conjugation experiment was carried out to transfer Δ*mcr-1* PN23 (IncX4) plasmid into *E. coli* J53.

## Conjugation experiments

Conjugation experiments were carried out in LB broth medium at 37°C using *E. coli* strain J53 as the recipient and MCR-1-positive *E. coli* (MCRPEC) natural strains as the donor. The overnight grown cultures of both the donor and recipient strain were washed with fresh LB medium and mixed at a 1:1 ratio. The mixed culture was incubated at 37°C overnight without shaking. Transconjugants were selected on LB agar containing 150 µg/ml sodium azide and 2 µg/ml colistin. In the case of mcr-knockout plasmid mutant (Δ*mcr-1* PN23 IncX4), *E. coli* MG1655 was used as the donor and the transconjugants were selected on 150 µg/ml sodium azide and 100 µg/ml ampicillin. The presence of plasmids in transconjugants was confirmed by PCR.

## Construction of pSEVA:MCR-MOR plasmid

*Moraxella* species have been identified as potential sources of MCR-1 (*Sun et al., 2018*; *Gao et al., 2016*). To study the *Moraxella* version of MCR (MCR-MOR), we custom-synthesized (Twist Bioscience) MCR-MOR gene (*Moraxella osloensis*, GenBank: AXE82_07515) and cloned this gene into pSEVA121 plasmid using Gibson assembly method. For cloning, the MCR-MOR fragment (insert DNA 1709 bp) and pSEVA backbone (vector DNA 4001 bp) containing ampicillin resistance marker were amplified by PCR with corresponding primers (*Supplementary file 4*) in Q5 High-Fidelity 2X Master Mix (New England BioLabs). Both the insert (MCR-MOR) and vector fragments were gel-purified using GeneJET Gel Extraction and DNA Cleanup Micro Kit (Thermo Scientific). The gel-purified PCR products were assembled together using NEBuilder HiFi DNA Assembly Master Mix (New England BioLabs) according to the manufacturer's instructions. Following the assembly, 2 ul of the assembly mixture was transformed into *E. coli* strain J53 and transformants were selected on LB agar containing 100 µg/ml ampicillin. The assembly of pSEVA MCR-MOR plasmid was verified by PCR.

## Physicochemical properties of AMPs

Protein amino acid frequencies and the fraction of polar and non-polar amino acids were counted with an in-house R script. PepCalc (Innovagen) calculator was used to calculate the net charge. Isoelectric point and hydrophobicity were calculated using Peptide Analyzing Tool (Thermo Scientific). Percentage of the disordered region, beta-strand region, coiled structure, and alpha-helical region was calculated with Pasta 2.0. The ExPasy ProtParam tool was used for calculating aliphatic index and hydropathicity. Aggregation hotspots were calculated by AggreScan.

## Determination of MIC

MICs were determined with a standard serial broth dilution technique with a minor modification that we previously optimized for AMPs (*Kintses et al., 2019*). Specifically, smaller AMP concentration steps were used (typically 1.2–1.5-fold) because AMPs have steeper dose–response curves than

standard antibiotics (*Yu et al., 2018*; *Lazzaro et al., 2020*), and therefore bigger concentration steps (such as twofold dilutions) cannot capture 90% growth inhibitions (i.e., MIC). 10-steps serial dilution was prepared in fresh MHB medium in 96-well microtiter plates where AMP was represented in nine different concentrations. Three wells contained only medium to monitor the growth in the absence of AMP. Bacterial strains were grown in MHB medium supplemented with appropriate antibiotic (100 μg/ml ampicillin for *E. coli* pSEVA MCR-1 and 1 μg/ml colistin for MCR natural plasmid) at 30°C overnight. Following overnight incubation, approximately $5 \times 10^5$ cells were inoculated into the wells of the 96-well microtiter plate. We used three independent replicates for each strain and the corresponding control. The top and bottom rows in the 96-well plate were filled with MHB medium to obtain the background OD value of the medium. Plates were incubated at 30°C with continuous shaking at 250 rpm. After 20–24 hr of incubation, $OD_{600}$ values were measured in a microplate reader (Biotek Synergy 2). After background subtraction, MIC was defined as the lowest concentration of AMP where the $OD_{600} < 0.05$. Bacterial susceptibility to human serum was also measured using the similar MIC assay described above. Human serum was purchased from Sigma.

## Membrane surface charge measurement

To measure bacterial membrane surface charge, we carried out a fluorescein isothiocyanate-labeled poly-L-lysine (FITC-PLL) (Sigma) binding assay. FITC-PLL is a polycationic molecule that binds to an anionic lipid membrane in a charge-dependent manner and is used to investigate the interaction between cationic peptides and charged lipid bilayer membranes (*Rossetti et al., 2004*). The assay was performed as previously described (*Spohn et al., 2019*; *Kintses et al., 2019*). Briefly, bacterial cells were grown overnight in MHB medium, centrifuged, and washed twice with 1× PBS buffer (pH 7.4). The washed bacterial cells were resuspended in 1× PBS buffer to a final $OD_{600}$ of 0.1. A freshly prepared FITC-PLL solution was added to the bacterial suspension at a final concentration of 6.5 μg/ml. The suspension was incubated at room temperature for 10 min and pelleted by centrifugation. The remaining amount of FITC-PLL in the supernatant was determined fluorometrically (excitation at 500 nm and emission at 530 nm) with or without bacterial exposure. The quantity of bound molecules was calculated from the difference between these values. A lower binding of FITC-PLL indicates a less net negative surface charge of the outer bacterial membrane.

## In vitro competition assay

To directly test the selective fitness benefits of MCR-1, we carried out in vitro competition experiment using a flow cytometry-based sensitive and reproducible method developed in our lab (*Yang et al., 2017*; *San Millan et al., 2016*; *Gifford et al., 2018*). Flow cytometry was performed on an Accuri C6 (Becton Dickinson, Biosciences, UK). We measured the competitive fitness of *E. coli* strain J53 harboring pSEVA MCR-1 in the absence and presence of an AMP. For this assay, we randomly selected five AMPs and colistin. *E. coli* harboring pSEVA plasmid without MCR-1 (called pSEVA <u>e</u>mpty <u>v</u>ector [EV]) was used as a control to calculate the relative fitness of *E. coli* pSEVA:MCR-1. These strains were competed against a GFP-labeled *E. coli* strain J53 to measure the relative fitness (see *Figure 1— figure supplement 2*). All competitions were carried out in MHB medium with six biological replicates per strain, as previously described (*Yang et al., 2017*; *San Millan et al., 2016*). Briefly, the bacterial cells were grown in MHB medium supplemented with 100 ug/ml ampicillin at 30°C overnight. The overnight grown cultures were washed with filtered PBS buffer to remove any antibiotic residues. The washed cells were diluted into a fresh MHB medium and mixed approximately at 1:1 ratio with GFP-labeled *E. coli* J53. Before starting the competition, the total cell density in the competition mix was around half million cells, as we also used for MIC assay. The initial density of fluorescent and nonfluorescent cells was estimated in the mix using medium flow rate, recoding 10,000 events, and discarding events with forward scatter (FSC) < 10,000 and side scatter (SSC) < 8000. After confirming the actual ratio close to 1:1, the competition plates were incubated at 30°C with shaking at 250 rpm. After overnight incubation, the competition mix was diluted in PBS buffer and cell densities were adjusted around 1000/μl. The final density of fluorescent and nonfluorescent cells was estimated in the competition mix. Using the initial and final density of fluorescent and nonfluorescent cells, the relative fitness was calculated as follows:

$$\text{Relative fitness} = \frac{\log_2\left(\frac{d1_{(\text{non}-\text{fluorescent})}}{d0_{(\text{non}-\text{fluorescent})}}\right)}{\log_2\left(\frac{d1_{(\text{fluorescent})}}{d0_{(\text{fluorescent})}}\right)},$$

where $d0$ and $d1$ represent cell density before and after the competition, respectively. Using this formula, the fitness of *E. coli* pSEVA:MCR-1 and *E. coli* pSEVA EV control was calculated (relative to GFP-labeled strain). In *Figure 1*, we expressed the fitness of *E. coli* pSEVA:MCR-1 strain relative to the control strain (i.e., *E. coli* pSEVA EV) and followed the procedure of error propagation to account for the uncertainty of the estimates:

$$\text{SE} = \sqrt{\left(\frac{\text{SD}_{\text{mcr1}}}{-f_{\text{mcr1}}}\right)^2 + \left(\frac{\text{SD}_{\text{EV}}}{-f_{\text{EV}}}\right)^2}$$

where $\bar{f}$ and SD are a mean estimate and its standard deviation for each corresponding strain based on six biological replicates. MCR1 and EV represent *E. coli* J53 carrying pSEVA:MCR-1 and *E. coli* J53 carrying empty vector control strain, respectively. See *Figure 1—figure supplement 2* for the gating strategy.

## In vivo virulence assay

Age and weight-defined TruLarv *G. mellonella* caterpillars were obtained in bulk from BioSystems Technology (Exeter, UK) and stored at 15°C in the absence of food. *E. coli* J53 pSEVA:MCR-1 and empty vector control strain was grown overnight in MHB broth and washed twice with sterile PBS. In the case of every experiment, treatment solutions were injected into the hemocoels of the larvae via the first right proleg using 10 μl Hamilton syringes (Reno, NV). Larvae were incubated in Petri dishes lined with filter paper at 37°C for 48 hr, and survival was documented every 6 hr. Insects were considered dead if they failed to respond to touch. Pretreatment was administered approximately 24 hr before bacterial injection, and in this time period the survival of the animals was not recorded. Before bacterial injection, the dead or sick animals were excluded from further experiments.

In order to establish the inoculum required to kill *G. mellonella* over 48 hr, 10 caterpillars were inoculated with 10 μl of bacterial suspensions containing $10^4$, $10^5$, $10^6$, $10^7$, and $10^8$ CFU/larva of *E. coli* strain carrying pSEVA empty vector control in PBS (data not shown). CFU number was verified by viable bacterial counts on MHB agar. Based on this preliminary experiment, $5 * 10^7$ and $1 * 10^8$ were determined as the ideal inoculum sizes to kill *G. mellonella* larvae.

LPS from pathogenic bacterial strain *E. coli* O111:B4 was purchased from Sigma-Aldrich (Merck KGaA, Darmstadt, Germany) that has been shown to stimulate host innate immunity (*Mukherjee et al., 2010*). LPS solutions from powder were prepared fresh by dissolving the powder in 1× PBS, and the solution was sterilized by heating at 80°C for at least 30 min. LPS pretreatment was administered similarly to bacterial treatment into the left first proleg approximately 24 hr before bacterial injection. In this time period, the survival of the animals was not continuously recorded. Before bacterial injection, the dead or sick animals were excluded from further experiments. In order to establish an ideal treatment dose of LPS, a dose–response experiment was performed with 1.25, 2.5, 5, 10, and 20 mg/ml LPS solution used for pretreatment (data not shown). Larvae were injected with 10 μl of each dose of LPS independently. In the case of animals injected with only LPS in the absence of bacteria, the survival of the animals was not affected, proving that LPS in itself has no significant toxic effects at the tested concentrations. In the case of injecting the animals with bacteria, LPS caused a very severe reaction and swift animal death. Because of that, the relatively small treatment dose of 2.5 mg/ml was chosen for the final experiment, and mock-treated larvae injected with PBS only were used an additional control (*Figure 4f*).

All experimental data were visualized with Kaplan–Meier survival curves, utilizing R packages *survival*, *survminer,* and *ggsurvplot*. p-Values in comparison of treatment groups within experiments were generated by these packages utilizing a standard log-rank test. p-Values comparing the results between experiments were obtained by comparing hazard ratios between the treatment lines based on the Cox proportional-hazards model.

## Acknowledgements

This work was supported by grants from the Wellcome Trust (106918/Z/15Z, CM), the Medical Research Council (MR/S013768/1, TW and CM), the National Natural Science Foundation of China (81861138051, YW), the European Research Council H2020-ERC-2014-CoG 648364-Resistance Evolution (CP), the European Research Council H2020-ERC-2019-PoC 862077- Aware, and Hungarian Academy of Sciences Momentum 'Célzott Lendület' Programme LP-2017-10/2017 (CP), National Research, Development and Innovation Office 'Élvonal' Programme KKP 126506 (CP), National Laboratories Program, National Laboratory of Biotechnology Grant NKFIH-871-3/2020, Gazdaságfejlesztési és Innovációs Operatív Program GINOP-2.3.2-15-2016-00014 (EVOMER, CP), Gazdaságfejlesztési és Innovációs Operatív Program GINOP-2.3.2-15-2016-00020 (MolMedEx TUMORDNS), and National Research, Development and Innovation Office, Hungary. LO was supported by the Biotechnology and Biological Sciences Research Council doctoral training partnership (BB/M011224/1). PS was supported by the ÚNKP-21-4-New National Excellence Program of the Ministry for Innovation and Technology from the source of the National Research, Development, and Innovation Fund. MC was supported by the Szeged Scientists Academy under the sponsorship of the Hungarian Ministry of Innovation and Technology (FEIF/433-4/2020-ITM_SZERZ). We thank Mei Li (Institute of Infection and Immunity, Cardiff University, UK) for assistance with the shipment of MCR-3 plasmids.

## Additional information

### Funding

| Funder | Grant reference number | Author |
| --- | --- | --- |
| Wellcome Trust | 106918/Z/15Z | Craig R MacLean |
| Medical Research Council | MR/S013768/1 | Craig R MacLean<br>Timothy R Walsh |
| National Natural Science Foundation of China | 81861138051 | Yang Wang |
| European Research Council | H2020-ERC-2014-CoG 648364 –Resistance Evolution | Csaba Pál |
| European Research Council | H2020-ERC-2019-PoC 862077–Aware | Csaba Pál |
| Hungarian Academy of Sciences | Momentum 'Célzott Lendület' Programme LP-2017-10/2017 | Csaba Pál |
| National Research, Development and Innovation Office | 'Élvonal' Programme KKP 126506 | Csaba Pál |
| National Laboratories Program, National Laboratory of Biotechnology Grant | NKFIH-871-3/2020 | Csaba Pál |
| Gazdaságfejlesztési és Innovációs Operatív Program | GINOP-2.3.2-15-2016-00014 | Csaba Pál |
| Gazdaságfejlesztési és Innovációs Operatív Program | GINOP-2.3.2-15-2016-00020 (MolMedEx TUMORDNS | Csaba Pál |
| Biotechnology and Biological Sciences Research Council | BB/M011224/1 | Liam P Shaw |
| Ministry for Innovation and Technology | ÚNKP-21-4-New | Petra Szili |

| Funder | Grant reference number | Author |
|---|---|---|
| Ministry for Innovation and Technology | FEIF/433-4/2020-ITM_SZERZ | Marton Czikkely |

The funders had no role in study design, data collection and interpretation, or the decision to submit the work for publication. For the purpose of Open Access, the authors have applied a CC BY public copyright license to any Author Accepted Manuscript version arising from this submission.

## Author contributions

Pramod K Jangir, Conceptualization, Data curation, Formal analysis, Validation, Investigation, Methodology, Writing – original draft, Project administration; Lois Ogunlana, Csaba Pál, Investigation, Methodology; Petra Szili, Marton Czikkely, Emily J Stevens, Investigation; Liam P Shaw, Formal analysis, Methodology; Yang Yu, Qiue Yang, Yang Wang, Timothy R Walsh, Resources; Craig R MacLean, Conceptualization, Resources, Formal analysis, Supervision, Funding acquisition, Methodology, Writing – original draft, Project administration

## Author ORCIDs

Pramod K Jangir ⬚ http://orcid.org/0000-0001-8330-0655
Csaba Pál ⬚ http://orcid.org/0000-0002-5187-9903

## Decision letter and Author response

Decision letter https://doi.org/10.7554/eLife.84395.sa1
Author response https://doi.org/10.7554/eLife.84395.sa2

# Additional files

## Supplementary files

• Supplementary file 1. List of the physicochemical and mechanistic properties of the antimicrobial peptides (AMPs) used in this study.
• Supplementary file 2. Competitive fitness data.
• Supplementary file 3. Antimicrobial susceptibility of MCR-*E. coli* to antimicrobial peptides (AMPs).
• Supplementary file 4. List of oligonucleotides used in this study.
• Supplementary file 5. List of bacterial strains used in this study.
• Supplementary file 6. Plasmids used in this study.
• MDAR checklist

## Data availability

All data generated or analysed during this study are included in this article and its supplementary Information.

The following previously published dataset was used:

| Author(s) | Year | Dataset title | Dataset URL | Database and Identifier |
|---|---|---|---|---|
| Shen C, Zhong L-L, Yu Y | 2020 | Dynamics of mcr-1 prevalence and mcr-1-positive *Escherichia coli* after the cessation of colistin use as a feed additive for animals in China: a prospective cross-sectional and whole genome sequencing-based molecular epidemiological study | https://www.ncbi.nlm.nih.gov/bioproject/PRJNA593695 | NCBI BioProject, PRJNA593695 |

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

## Appendix 1

The proportion of B2 isolates in a given category of isolates that are MCR-1-positive, $P\,(MCR|B2)$, depends on related probabilities as follows:

$$P\,(\text{MCR}|\text{B2}) = \frac{P\,(\text{B2}|\text{MCR}) \times P\,(\text{MCR})}{P\,(\text{B2})}$$

The Shen et al. dataset provides us with estimates for the two terms in the numerator on the right-hand side, but not the denominator.

For an association to exist between B2 and MCR-1 in infection compared to healthy carriage, this is equivalent to

$$P_i\,(\text{MCR}|\text{B2}) > P_h\,(\text{MCR}|\text{B2})$$

Substituting in values from the first equation, we obtain

$$P_i\,(\text{B2}) < \frac{P_i\,(\text{B2}|\text{MCR})\,P_i\,(\text{MCR})}{P_h\,(\text{B2}|\text{MCR})\,P_h\,(\text{MCR})} \times P_h\,(\text{B2})$$

The values from *Appendix 1—table 1* give the following point estimate (95% CI for factor: 0.5-Inf):

$$P_i\,(\text{B2}) < 2.4 \times P_h\,(\text{B2})$$

**Appendix 1—table 1.** Values from the Shen dataset for the relevant proportions.

| | MCR-1 (PCR) | | B2 and MCR-1 (whole genome sequencing [WGS] of MCR-positive) | | B2 proportion |
|---|---|---|---|---|---|
| Category | n | P (MCR) | n | P (B2|MCR) | P (B2) |
| Patients with infection | 59/3724 | 2 (1–2) | 18/55 | 33 (20–45) | $P_i$(B2) |
| Patients with colonization | 364/2395 | 15 (14–17) | 4/110 | 4 (0–7) | $P_c$(B2) |
| Healthy carriage | 353/3422 | 10 (9–12) | 3/144 | 2 (0–4) | $P_h$(B2) |

MCR: mobile colistin resistance.

The overall proportion of B2 in each category is therefore crucial for evidence of an association. This data is not available for the Shen dataset. By way of an example, if we take $P_h\,(B2) \approx 25\%$ then we would have that an association between B2 and MCR-1 in infection requires

$$P_i\,(B2) < 60\%$$

We searched for possible values for $P\,(B2)$ in isolates from healthy people compared to infection isolates. *Appendix 1—table 2* shows published values from a variety of settings to indicate the possible range. While $P_i\,(B2)$ appears to be above 60% in England, published values from China appear to be lower than 60%, thus making it possible that an association between MCR-1 and B2 may exist given the Shen data. Similarly, $P_h\,(B2)$ was as high as 48% in one study from China. Given this variability, it is not possible to reach a conclusion about evidence of association between B2 and MCR-1 without reliable data on $P\,(B2)$ for the same setting.

**Appendix 1—table 2.** Values identified in the literature for $P(B2)$ in healthy people (h) compared to infections (i) in different settings.

| Study | Details | P(B2) (*i*: infection; *h*: healthy) |
|---|---|---|
| *Tenaillon et al., 2010* | 2010 synthesis of previous data on phylogroup prevalence in healthy carriage (n = 1117 subjects) | $P_h(B2) = 26\%$ |
| *Luo et al., 2011* | Healthy food handlers in hospital in Beijing, China 2009 (n = 92) | $P_h(B2) = 48\%$ |
| *Marin et al., 2022* | Healthy people, France 1980–2010 (n = 436) | $P_h(B2) = 29\%$ |
| *Li et al., 2010* | Healthy people in Fujian, China 2010 (n = 325) | $P_h(B2) = 16\%$ |
| *Petitjean et al., 2021* | Analysis of *E. coli* genomes in EnteroBase (n = 70,301) | $P(B2) = 18\%$* |
| *Kallonen et al., 2017* | Bloodstream infection isolates in England, 2001–2012 (n = 1509) | $P_i(B2) = 67\%$ |
| *Davies et al., 2020* | Bloodstream infection isolates in England, 2013–2015 (n = 976) | $P_i(B2) = 66\%$ |
| *Zhang et al., 2021* | Bloodstream infection isolates in Shanxi, China, 2019–2020 (n = 76) | $P_i(B2) = 34\%$ |
| *Hu et al., 2013* | CTX-M-producing *E. coli* in Hangzhou, China 2010–2012 (n = 46 healthy, n = 36 clinical) | $P_h(B2) = 18\%$ $P_i(B2) = 39\%$ |
| *Rodríguez et al., 2021* | Bloodstream infection isolates in Spain, 1996–2016 (n = 649 isolates representing 7165 infection episodes) | $P_i(B2) = 53\%$ |

*Not separated by category since contains a mixture of isolate sources.

