## [Editor Report]

Antimicrobial peptides (AMPs) are a class of antibiotics that are inspired by natural components of innate immunity, which raises the specter of bacteria becoming resistant to both. Jangir et al. test this idea and find compelling evidence that a plasmid that encodes resistance to the AMP colistin also increases resistance to AMPs produced by humans, pigs, and chickens, enables the bacteria to grow better in low levels of AMP, and increases bacterial virulence in an insect model of infection. This important study will be of interest to both evolutionary biologists and microbiologists focused on antimicrobial therapy and suggests that the evolution of resistance to these compounds can have collateral effects on immune evasion as well.

---

## [Decision Letter]

**Decision letter after peer review:**

Thank you for submitting your article "The evolution of colistin resistance increases bacterial resistance to host antimicrobial peptides and virulence" for consideration by *eLife*. Your article has been reviewed by 2 peer reviewers, and the evaluation has been overseen by a Reviewing Editor and George Perry as the Senior Editor. The following individual involved in the review of your submission has agreed to reveal their identity: Camilo Barbosa (Reviewer #1).

Essential revisions:

1) Supp Figure 6 should be in the main text and more clearly explained, including some explicit experimental predictions of the model.

2) Claims regarding the fitness effects of these adaptations should be toned down given that only a single host strain is used. The claims for selective benefits of AMP adaptations would be stronger (and more sensitive) if the authors conducted fitness assays that could unambiguously assess the effects of MCR in serum and/or Galleria. Serum is preferable because Galleria experiments are a black box because bacterial numbers aren't measured. The assumption is that Galleria AMPS somehow alters the bacterial clearance of the MCR strain, but there's no evidence of this. The authors should at least note the different possibilities.

*Reviewer #1 (Recommendations for the authors):*

– I am not sure what the constitutive levels of AMPs in serum are, but perhaps an easy way to complement these results to support the conclusion that MCR-1 mediated colistin resistance is beneficial against cocktails of AMPs in human serum would be to have a mixed treatment with all evaluated AMPs from the same 'type' of host.

– I understand that the range of values in Figures 2a and b are quite different, but it would be simpler to keep the results for COL in the same panel as Figure 2b. Not too relevant, just a suggestion.

– I think the results of Sup. Figure 6 is very important and could be presented within the main manuscript.

*Reviewer #2 (Recommendations for the authors):*

It would be helpful if the y-axis, "Relative change in MIC", was on the same scale in Figures 2 and 3.

---

## [Author Response]

Essential revisions:(1) Supp Figure 6 should be in the main text and more clearly explained, including some explicit experimental predictions of the model.

Thank you for the comment. Following this, we have moved supplementary Figure 6 to the main text (Figure 4c and 4d) and have revised this section to make the prediction more explicit (lines 245-257, page 10).

(2) Claims regarding the fitness effects of these adaptations should be toned down given that only a single host strain is used. The claims for selective benefits of AMP adaptations would be stronger (and more sensitive) if the authors conducted fitness assays that could unambiguously assess the effects of MCR in serum and/or Galleria. Serum is preferable because Galleria experiments are a black box because bacterial numbers aren't measured. The assumption is that Galleria AMPS somehow alters the bacterial clearance of the MCR strain, but there's no evidence of this. The authors should at least note the different possibilities.

We agree with the reviewer about the limitations of these assays and thus have re-written the discussion to better reflect this (lines 370-382, page 15)

Reviewer #1 (Recommendations for the authors):– I am not sure what the constitutive levels of AMPs in serum are, but perhaps an easy way to complement these results to support the conclusion that MCR-1 mediated colistin resistance is beneficial against cocktails of AMPs in human serum would be to have a mixed treatment with all evaluated AMPs from the same 'type' of host.

Thank you for the suggestion. Although determining the exact serum levels of host AMPs is challenging and it varies considerably between individuals, some studies have measured it in healthy and diseased conditions. For example, serum levels of LL37 in healthy people range from 0.25-12ng/ml, whereas in the case of bacterial infections it can reach up to 80-90ng/ml (PMID: 30799994, 25197165). It would be interesting to test the impact of MCR-1 on resistance to mixtures of host AMPs, but we have decided not to carry out further experiments with AMP cocktails as it is generally unclear to what extent these AMPs from the same animal species are co-expressed in host tissues, as highlighted by Table 1.

We think that our results from human serum assay and in vivo infection point to an immunologically relevant and complex host environment where bacterial cells encounter not only the host antimicrobial peptides but also other antimicrobials, such as complement systems and antimicrobial proteins in human serum, and the host immune system, and growth environment/nutrients in Galleria model. Moreover, testing MCR-1 in LPS-treated Galleria (Figure 4f) reflects the condition of a high level of mixed AMPs as LPS treatment induces innate immunity and the expression of diverse immunity peptides in Galleria (PMID: 19897755).

– I understand that the range of values in Figures 2a and b are quite different, but it would be simpler to keep the results for COL in the same panel as Figure 2b. Not too relevant, just a suggestion.

As highlighted in the comment, we think that it is better to keep them separately due to big differences in MICs.

– I think the results of Sup. Figure 6 is very important and could be presented within the main manuscript.

Thank you for the suggestion, we have moved it to the main text.

Reviewer #2 (Recommendations for the authors):It would be helpful if the y-axis, "Relative change in MIC", was on the same scale in Figures 2 and 3.

Done.